# StarGAN-VC++: Towards Emotion Preserving Voice Conversion Using Deep Embeddings

*Arnab Das[1,3*], Suhita Ghosh[1*], Tim Polzehl[3], Ingo Siegert[2], Sebastian Stober[1]*

[1]Artificial Intelligence Lab (AILab), Otto-von-Guericke-University, Magdeburg, Germany
[2]Mobile Dialog Systems, Otto-von-Guericke-University, Magdeburg, Germany
[3]Speech and Language Technology, German Research Center for Artificial Intelligence (DFKI)

{suhita.ghosh,ingo.siegert,stober}@ovgu.de
{arnab.das,tim.polzehl}@dfki.de

## Abstract

Voice conversion (VC) transforms an utterance to sound like another person without changing the linguistic content. A recently proposed generative adversarial network-based VC method, StarGANv2-VC is very successful in generating natural-sounding conversions. However, the method fails to preserve the emotion of the source speaker in the converted samples. Emotion preservation is necessary for natural human-computer interaction. In this paper, we show that StarGANv2-VC fails to disentangle the speaker and emotion representations, pertinent to preserve emotion. Specifically, there is an emotion leakage from the reference audio used to capture the speaker embeddings while training. To counter the problem, we propose novel emotion-aware losses and an unsupervised method which exploits emotion supervision through latent emotion representations. The objective and subjective evaluations prove the efficacy of the proposed strategy over diverse datasets, emotions, gender, etc.

**Index Terms**: voice conversion, emotion preservation, StarGAN

## 1. Introduction

Voice conversion (VC) is a technique that transforms the speech of one speaker to make it sound like another speaker's voice, while keeping the linguistic content intact and ensuring that the quality, naturalness and comprehensibility of the converted speech remain high [1]. VC systems have numerous practical use cases in various domains [2]. For example, in speech therapy, a VC system could be used to record sessions for later analysis while ensuring that patient confidentiality is maintained, a mandatory request resulting from the requirement to adhere to the guidelines of the General Data Protection Regulation (GDPR) [3]. VC systems could also be useful in the entertainment industry to perform tasks like voice dubbing. Furthermore, preserving the emotional state of the speaker parallel to providing anonymized speech is necessary in all VC applications where the emotional information is necessary for further processing, such as when analyzing the speech for mental health issues or interaction with an affect-aware avatar. Therefore, VC methods should ensure that the original speaker's emotional state is not lost during the conversion process.

Many deep learning-based approaches have been proposed for voice conversion [4]. Early deep neural network (DNN)-based methods [5] largely concentrated on the concept of frame-wise spectral feature conversion. Soon after, long short-term memory (LSTM) based sequence-to-sequence models [6] were employed for the task and produced high-quality converted samples. Although the converted samples generated by sequence-to-sequence models are more natural, they suffer from mispronunciation and training instability [7]. Moreover, these methods need numerous parallel

utterances to learn [8], which is a very expensive, time-consuming, and burdensome task in itself.

To alleviate the issue of obtaining expensive parallel training data, several non-parallel-based DNN methods were proposed, which are based on variational autoencoder (VAE) [9, 10, 11] and cycle-consistent generative adversarial network (Cycle-GAN) [12, 13, 14]. The VC methods utilizing VAE typically attempt to extract disentangled speaker and content embeddings from utterances through a reconstruction loss. The VAE-based approaches produce over-smoothed conversions, which leads to a poor-quality, buzzy-sounding speech [7, 8]. The CycleGAN-based frameworks use cycle consistency loss, which learns both forward and reverse conversion of samples between two speakers using non-parallel training utterances. A major problem with CycleGAN-based frameworks is that they require training of one generator for each pair of speakers, which makes it impractical for many-to-many VC use cases. The CycleGAN-based frameworks are also criticized for producing low-quality converted samples [7].

A few recent text-to-speech (TTS) based voice conversion methods [15] extract the content through two modules, automatic speech recognition (ASR) and TTS. The linguistic content is first extracted from the source speech through an ASR. Further, a TTS is fed with the ASR generated transcription and a target speaker's embedding to generate the converted utterance. The naturalness and intelligibility of the converted speech using such ASR and TTS-based systems are typically high [16]. However, they fail to preserve the prosody or affective state of the source speaker, as they use only the linguistic content of the source utterance to generate the converted speech. Further, the performance of such VC systems is dependent on the quality of transcriptions produced by the ASR.

Recently, several StarGAN-based [17] non-parallel many-to-many VC frameworks [18, 19, 20] have been proposed. Among those, the StarGANv2-VC [20] framework is especially interesting as it generates fundamental frequency (F0) consistent, natural sounding and highly intelligible converted samples [21]. Moreover, the architecture design makes the framework very scalable, making it suitable for utterances of any length and, its fast conversion capability makes it suitable for real-time applications. However, the model fails to preserve the affective state of the source speaker when the source utterance has a large variation in the acoustic parameters.

In this paper, we investigate the reason for StarGANv2-VC's failure in preserving the source speaker's emotion in converted speech. We also propose a novel method to circumvent the said problem, by using an unsupervised emotion supervision technique through latent emotion representations. Further, we propose losses which prevent emotion leakage from the reference audio used to generate the speaker embeddings, which also leads to a better disentanglement of speaker and emotion representations. We evaluate the proposed method extensively with three datasets for various emotions, different gender, and accent groups. The objective

---

*These authors contributed equally to this work

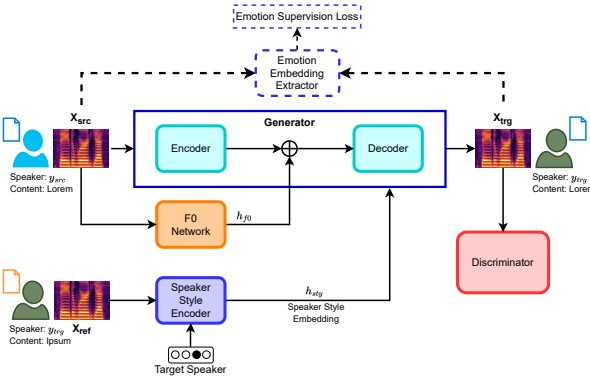

Figure 1: *StarGANv2-VC architecture for VC. The dashed parts belong to the proposed emotion supervision method.*

and subjective evaluation results depict that the proposed method significantly improves the emotion preservation capability over vanilla StarGANv2-VC for all cases.

## 2. StarGANv2-VC and Emotion Leakage

### 2.1. Architecture

The architecture diagram of StarGANv2-VC [20] is presented in Figure 1, along with the proposed emotion supervision components. The generator G produces the converted utterance $X_{trg}$, which comprises an encoder (EN) and a decoder. The generator is fed with an utterance $X_{src}$ belonging to the source speaker $y_{src}$ and a speaker style-embedding $h_{sty}$ belonging to the target speaker $y_{trg}$, where $y_{src}, y_{trg} \in Y$. As described by the authors, the generator also consumes the source F0 embeddings $h_{f0}$ produced by a pre-trained network, which enables the model to produce F0 consistent conversions [20]. The speaker-style-encoder (SE) generates the speaker-style-embedding $h_{sty}$ from a randomly selected reference utterance $X_{ref}$ of the target speaker $y_{trg}$, given the target speaker-code. The embedding $h_{sty}$ represents speaker-style information, such as accent. Hence, the converted sample $X_{trg} = G(X_{src}, h_{f0}, h_{sty})$, contains the speaker characteristics of the target speaker, but the intonation and linguistic content of the source utterance. To encourage G in producing unique samples, a separate mapping network (M) is also trained along with SE, as done in [20]. During training, speaker-style embeddings are generated using both SE or M alternatively in each optimization step. The module M produces target speaker-embedding from a random latent vector sampled from a normal Gaussian distribution and target speaker-code. The discriminator module consists of two adversarial classifiers: a quality classifier (C) and a speaker classifier (C$_{sp}$). The C classifier classifies the real and fake samples conditioned on the speaker-code. The classifier (C$_{sp}$) classifies the source speaker of the converted sample during the discriminator training phase and classifies the target speaker during the generator training phase, as proposed in [20]. This further encourages G to suppress the source speaker's traits in converted samples.

### 2.2. Emotion Leakage by Speaker Embedding

The speaker-style-encoder (SE) extracts the target speaker's style-embedding from the reference utterance, shown as $X_{ref}$ in Figure 1. For the emotion-leakage introspection, we train a vanilla StarGANv2-VC with English utterances from Emotional Speech Database (ESD) corpus [22], which has emotional utterances. After the training, we extract the speaker embeddings. For illustration, we

choose two different speakers, 0012 (male) and 0016 (female) from ESD. The model is trained using utterances from these speakers. The speaker embeddings are projected onto a 2D space using tSNE transformation, and the results are presented in Figure 2. The speaker embeddings for utterances from a single speaker should conform to a compact region in space, regardless of the emotion of the utterances. On the contrary, Figure 2 reveals that the embeddings form unnecessary grouping based on emotion. This shows that the SE fails to disentangle emotional cues from the reference utterances when it generates target speaker embedding from those utterances. Consequently, this unintended emotional information leaks to the decoder along with the target speaker's style representation. This confuses the decoder while generating the utterance using the target speaker's voice. This leakage occurs due to two reasons: (i) the absence of a training objective that encourages the SE to perform the disentanglement between speaker-dependent and speaker-independent features. (b) the speaker-style reconstruction loss $\mathcal{L}_{style}$ [20] used in the vanilla StarGANv2-VC, shown in Eqn. 1.

$$\mathcal{L}_{style} = \big[ |SE(X_{ref}, y_{trg}) - SE(X_{trg}, y_{trg})| \big] \tag{1}$$

The speaker-style reconstruction loss ensures that the style embeddings can be regenerated from the generated samples. This loss intends to bring the speaking style of the converted sample closer to the reference sample, as both of them belong to the same (target) speaker. This further exacerbates the domain leakage problem, as the target speaker-embedding is not disentangled from the emotional cues of the reference utterance. Therefore, by minimizing this loss, the converted speech tends to be closer to the reference also in terms of emotion, which subdues the emotion of the source speech. This cripples the emotion preservation capability of the vanilla StarGANv2-VC.

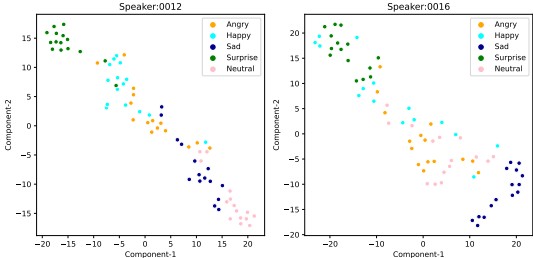

Figure 2: *2D tSNE plot for speaker embedding generated by the style encoder of the vanilla StarGANv2-VC.*

## 3. Methods

The availability of a large amount of good quality emotion labels is difficult. Further, in non-parallel voice conversion, the emotion label of the converted samples is not available. Therefore, to circumvent the emotion leakage problem, we propose an unsupervised emotion supervision technique using emotion representations. The emotion representations are deep emotion-embeddings, which contain the information about the affective state of the utterance. The proposed method encourages the generator to preserve the affective state of the source in the converted samples.

### 3.1. Deep Emotion Embedding and Emotion Supervision

One way of providing emotion supervision is to extract latent emotion representations from the source and the converted samples, and then minimize the distance between them. To this end, an emotion-embedding extraction network is needed

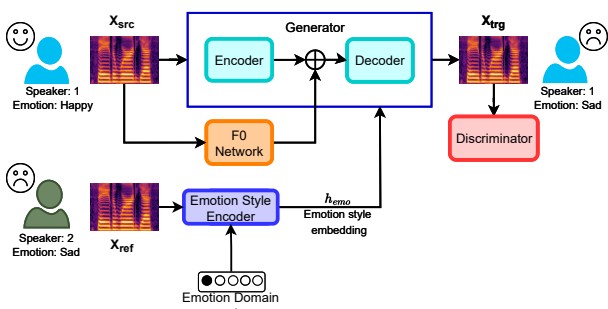

Figure 3: *StarGANv2-VC used in emotion conversion. For example, a happy utterance from speaker 1 is converted to a sad utterance. The emotion-style encoder generates sad emotion embedding from a sad reference utterance of speaker 2.*

to produce the latent emotion representations directly from the utterances. A two-stage training approach is proposed to create this emotion-embedding extraction network.

At Stage-I, an emotion conversion framework is trained, where the emotion is converted instead of speaker-traits. We used the vanilla StarGANv2-VC framework for the emotion conversion task, as shown in Figure 3. In this case, the style encoder captures the representations of emotion instead of speaker. The working principle of the style encoder in the emotion conversion task is depicted in Figure 4. The shallow shared convolution layers extract a 512 dimensional shared latent representation from the reference mel-spectrogram. Consequently, the fully connected (FC) layers project this shared embedding to emotion-specific 64 dimensional emotion-style embeddings, $h_{emo1}, ..., h_{emoN}$, where $N$ is the number of emotion classes. Finally, an emotion style-embedding $h_{emo}$ corresponding to the utterance needs to be selected based on the emotion-code $e_{trg}$. The emotion-code denotes the emotion class of the utterance. Therefore, when the emotion ground truth is not available, the selection of the emotion-code is not possible. This makes the current technique unusable for emotion extraction for VC, as the converted samples do not have emotion ground truth. Therefore, we need a mechanism which does not depend on the availability of emotion ground truth, which is achieved at Stage-II.

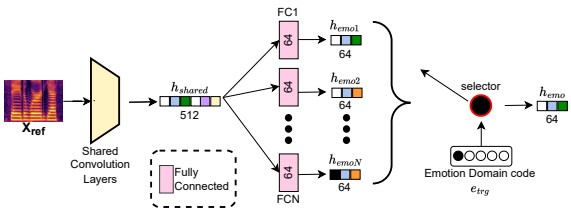

Figure 4: *Architecture and the working mechanism of the style encoder for emotion conversion.*

At Stage-II, the linear projecting FC layers are removed from the pre-trained SE and a classification head consisting of three FC layers is placed on top of the shared convolution layers, as shown in Figure 5. The classifier is then trained for a supervised emotion classification task. During training, the weights for the shallow pre-trained convolution layers remain fixed and only the weights for the classification head are optimized. After Stage-II training, the 64 dimensional output features of the second FC layer of the classification head can be used as the latent emotion representation. Now, this

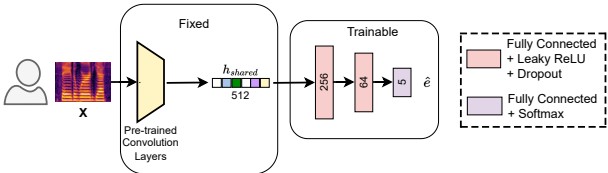

Figure 5: *Block diagram of the emotion classifier. $X$ is any input mel-spectrogram. The trainable emotion classification head is added on top of pre-trained shared convolution layers. The classification head predicts the emotion class $\hat{e}$.*

network serves as a differentiable black-box emotion-embedding extractor module $C_{emo}$ for the VC task, as shown in Figure 1. The module can be used to extract deep emotion representations from any utterance, irrespective of the presence of emotion ground truth. A loss is computed by equating the emotion representations from the source and converted, as shown in Eqn. 2.

$$\mathcal{L}_{demo} = \mathbb{E}_{X_{src}, X_{trg}} \big[ |C_{emo}(X_{src}) - C_{emo}(X_{trg})| \big] \quad (2)$$

We minimize this additional $\mathcal{L}_{demo}$ loss while training the StarGANv2-VC for the VC task, which encourages emotion preservation and suppress any emotion leakage from the reference utterance.

### 3.2. Style Reconstruction Loss

To encourage the disentanglement of the target speaker's style-embedding from any unintended emotion-indicating information of the reference utterance, we augment the style reconstruction loss $\mathcal{L}_{style}$ as shown in Eqn. 3. Here, $X_{ref2}$ is another randomly selected reference utterance for the same target speaker. The loss facilitates the model to bring the target speaker style embeddings extracted from $X_{ref}$ and $X_{ref2}$ closer.

$$\begin{aligned}
\mathcal{L}_{style} = & \big[ |SE(X_{ref}, y_{trg}) - SE(X_{trg}, y_{trg})| \big] \\
& + \big[ |SE(X_{ref2}, y_{trg}) - SE(X_{trg}, y_{trg})| \big] \\
& + \big[ |SE(X_{ref}, y_{trg}) - SE(X_{ref2}, y_{trg})| \big] \quad (3)
\end{aligned}$$

### 3.3. Conversion Invariant Feature Preservation Loss

In vanilla StarGANv2-VC, all the losses are applied directly to the generator output, and gradients are back-propagated all the way through the decoder and encoder. Consequently, the shallow convolution layers of the encoder get very little supervision about the conversion invariant features. These conversion invariant features, such as content and emotion-related features, need to be preserved as much as possible in the latent output of the encoder. To this end, we propose a new loss $\mathcal{L}_{inv}$ as shown in Eqn. 4, applied directly to the encoder EN. The loss minimizes the distance between the latent code generated from the source and the converted sample, forcing the encoder to preserve more content and emotion-related features in its latent output. This loss further helps in disentanglement of content features from speaker-specific features.

$$\mathcal{L}_{inv} = \mathbb{E}_{X_{src}, X_{trg}} \big[ |EN(X_{src}) - EN(X_{trg})| \big] \quad (4)$$

### 3.4. Overall Training Objective

The overall training objective for the generator is presented in Eqn. 5, where $\mathcal{L}_{demo}$ and $\mathcal{L}_{inv}$ are the proposed losses, and the rest are taken from [20]. $\mathcal{L}_{adv}$ is the typical adversarial loss. $\mathcal{L}_{spk}$ is adversarial speaker classifier loss. $\mathcal{L}_{div}$ encourages the generator to

produce diversified samples when given different style-embeddings. $\mathcal{L}_{asr}$ is linguistic content preservation loss. $\mathcal{L}_{norm}$ is the norm consistency loss which ensures the preservation of voiced/unvoiced intervals. $\lambda_{cycle}$ is cycle consistency loss to encourage the generator to learn a bijective mapping between source and target speakers and, $\mathcal{L}_{F0}$ is F0 consistency loss helps in generating F0-consistent samples.

$$
\begin{aligned}
\min_{G,SE,M} \quad & \mathcal{L}_{adv} + \lambda_{spk}\mathcal{L}_{spk} + \lambda_{style}\mathcal{L}_{style} - \lambda_{div}\mathcal{L}_{div} \\
& + \lambda_{asr}\mathcal{L}_{asr} + \lambda_{norm}\mathcal{L}_{norm} + \lambda_{cycle}\mathcal{L}_{cycle} \\
& + \lambda_{F0}\mathcal{L}_{F0} + \lambda_{demo}\mathcal{L}_{demo} + \lambda_{inv}\mathcal{L}_{inv}
\end{aligned} \tag{5}
$$

The training objective for the adversarial classifiers is presented in Eqn. 6. The adversarial quality classifier maximizes the adversarial loss $\mathcal{L}_{adv}$. The speaker classifier minimizes $\mathcal{L}_{aspk}$ through the classification of the source speaker. Each loss is weighted with a corresponding $\lambda$ hyperparameter.

$$
\min_{C,C_{sp}} \quad -\mathcal{L}_{adv} + \lambda_{aspk}\mathcal{L}_{aspk} \tag{6}
$$

# 4. Datasets, Experiments and Results

## 4.1. Training Details

We term our approach as StarGAN-VC++ and the vanilla StarGANv2-VC framework as Baseline. We train our models with a random split of 0.8/0.1/0.1 for train/validation/test. All the utterances are re-sampled to 24 kHz. Each model is trained for 60 epochs on log mel-spectrograms with a batch size of 16 using a Tesla V100 (32 GB) GPU, with a training time of around 26 hours. To train our model we set, $\lambda_{spk} = 0.5$, $\lambda_{aspk} = 0.1$, $\lambda_{style} = 1$, $\lambda_{div} = 1$, $\lambda_{asr} = 10$, $\lambda_{norm} = 1$, $\lambda_{cycle} = 5$, $\lambda_{F0} = 5$, $\lambda_{demo} = 2$, and $\lambda_{inv} = 5$. AdamW [23] is used with the learning rate of $10^{-4}$. A HiFiGAN [24] vocoder is trained with datasets mentioned in Table 1. The vocoder generates a 1 minute long waveform from the converted mel-spectrogram in 0.1 seconds. For the assessment of emotion preservation, we train a support vector machine (SVM) based emotion classifier as done in [25].

## 4.2. Datasets

Four datasets have been used in this work, which have been used for different purposes, as shown in Table 1. We consider 5 emotion classes $e \in \{$happy, sad, anger, neutral, surprise$\}$ for the emotion preservation evaluation.

Table 1: *Depicts usage of datasets for different purposes. The column Emotion denotes whether the dataset is used to train the emotion conversion network and, the emotion classifier for evaluation.*

| Dataset | VC Training | VC Evaluation | Vocoder Training | Emotion |
|---|---|---|---|---|
| ESD | Yes | Yes | Yes | Yes |
| VCTK | Yes | Yes | Yes | No |
| RAVDESS | No | No | Yes | Yes |
| CREMA-D | No | No | Yes | No |

**Emotional Speech Database (ESD)** [22]: The corpus contains English and Chinese emotional utterances belonging to the 5 emotion classes in $e$. We consider only English utterances in our work, where 350 utterances per emotion class are spoken by 5 male and 5 female native English speakers.

**Voice Cloning Toolkit (VCTK)** [26]: The dataset contains English utterances from 109 speakers having various accents. For training of VC models, we consider utterances having English accent from 5 randomly selected males and females. The evaluation is performed on utterances having English, American and Canadian accents.

**Ryerson Audio-Visual Database of Emotional Speech and Song (RAVDESS)** [27]: This corpus comprises 2880 utterances spoken by 24 professional actors from North America. The dataset has ground truth for 7 emotion classes, but we consider the ones mentioned in $e$.

**Crowd-sourced Emotional Multimodal Actors Dataset (CREMA-D)** [28]: This is another corpus having emotional utterances having the mentioned 5 emotion classes in $e$. The dataset contains 7442 clips from 91 actors belonging to diverse age groups and ethnicity.

## 4.3. Evaluation Setup

To evaluate the efficacy of the proposed methods, we perform both objective and subjective evaluations. For all assessments, we choose an equal number of male and female source and target speakers from each dataset.

**Objective Evaluation**: We choose test utterances from 10 source speakers and 6 target speakers from the ESD dataset, leading to 2100 ESD→ESD conversions. For VCTK→VCTK, we form 2000 conversions by selecting test utterances from 10 source speakers and 6 target speakers.

To evaluate emotion preservation, we compare the models using 3 metrics: i) $ACC_{gt}$: accuracy between the source emotion ground truth and the SVM predicted emotion label of the converted utterances. ii) $ACC_{svm}$: accuracy between the predicted labels of the source and the converted samples, where both are predicted by the SVM. The metric is especially useful when no emotion ground truth is available, as in VCTK dataset. 3) $MAE_{embed}$: mean absolute error (MAE) between the emotion-embedding of the source and converted utterances, where the emotion embedding is extracted using the automatic embedding extractor. We report the pitch correlation coefficient ($PCC$) [29] as a measure of intonation preservation, predicted mean opinion score ($pMOS$) [30] as a measure of the quality of converted samples, and character error rate ($CER$) as a content preservation evaluation measure. As a measure of speaker anonymization, we report speaker similarity score ($SSS$) between the source and the converted speech. The score is generated using a speaker verification toolkit [31] from Hugging-Face repository[1].

**Subjective Evaluation**: We randomly choose 100 conversions for the subjective evaluation, as it is expensive and time-consuming to assess all of them. A user study was conducted on Crowdee[2] platform, where 200 native English speakers participated. Each subject performs two types of assessments: 1) verify emotion leakage from reference: a triplet of converted, source and reference utterances were provided. The subjects were asked to select between source and reference utterances, the one which has a similar emotion as the converted utterance (ignoring content or voice similarity). ii) voice quality: assess the naturalness on a 5-point scale (1: bad to 5: excellent). The users were not apprised about whether the converted utterance is produced by Baseline or by the proposed model. Each of the tasks was rated by at least 5 subjects. The subjects were provided with an anchor question and also hidden trapping questions for quality check. The subjects failing the trapping questions were not considered for the analysis.

Table 2: *Objective evaluation results. Mean and standard deviation (in brackets) are reported. 'All Conv.' includes all conversions. Type column denotes different conversion sub-groups, such as source-emotion, source → target accents, source → target genders (M→M, M→F, F→F, F→M) or, 'All' indicates all sub-groups. StarGAN-VC++ is denoted by SG++.*

| Source - Target | Type | ACC$_{gt}$ [%] ↑ | | ACC$_{svm}$ [%] ↑ | | MAE$_{embed}$ [×10²]↓ | | PCC [×10²]↑ | | pMOS ↑ | | CER [%]↓ | | SSS [×10²]↓ | |
|---|---|---|---|---|---|---|---|---|---|---|---|---|---|---|---|
| | | Baseline | SG++ | Baseline | SG++ | Baseline | SG++ | Baseline | SG++ | Baseline | SG++ | Baseline | SG++ | Baseline | SG++ |
| **All Conv.** | All | 18.5 | **30.4** | 25.9 | **52.7** | 58.2 (19.2) | **45.7** (13.1) | 77.3 (15.3) | **78.3** (16.7) | 3.5 (0.5) | 3.5 (0.6) | 3.5 (8.5) | **3.2** (8.0) | 24.4 (15.9) | **23.9** (16.6) |
| **ESD → ESD** | All | 20.2 | **48.9** | 20.9 | **65.0** | 64.4 (22.6) | **40.8** (12.5) | 80.2 (13.4) | **84.2** (10.6) | 3.8 (0.4) | 3.8 (0.4) | 3.8 (8.4) | **3.0** (7.4) | **26.0** (18.8) | 28.0 (19.4) |
| | Happy | 17.2 | **47.0** | 19.1 | **54.7** | 62.0 (18.8) | **40.6** (11.5) | 78.8 (15.8) | **84.1** (11.6) | 3.6 (0.5) | **3.7** (0.4) | 5.9 (9.7) | **3.5** (8.5) | - | - |
| | Angry | 12.5 | **76.5** | 12.5 | **77.5** | 62.1 (18.8) | **35.3** (8.6) | 78.0 (15.3) | **84.2** (10.7) | **3.8** (0.4) | 3.7 (0.3) | 2.9 (6.3) | **2.2** (6.0) | - | - |
| | Sad | 12.6 | **45.4** | 12.6 | **45.1** | 62.9 (16.8) | **46.1** (13.4) | 85.0 (11.5) | **86.0** (11.2) | 3.8 (0.4) | **4.0** (0.4) | 2.7 (7.2) | **1.9** (5.2) | - | - |
| | Surprise | 0.0 | 0.0 | 1.5 | **62.9** | 83.5 (25.0) | **44.7** (14.7) | 80.7 (9.9) | **84.6** (7.9) | 3.7 (0.4) | 3.7 (0.4) | 6.0 (10.5) | **5.4** (9.3) | - | - |
| | Neutral | 79.7 | **90.7** | 79.7 | **90.7** | 45.9 (12.7) | **36.4** (8.4) | 78.4 (12.5) | **81.4** (11.1) | 4.1 (0.4) | **4.2** (0.4) | 1.4 (5.5) | **1.2** (5.6) | - | - |
| | M→M | 19.0 | **47.2** | 19.6 | **54.4** | 62.9 (19.1) | **40.3** (11.4) | 76.8 (13.7) | **82.4** (11.2) | 3.6 (0.4) | **3.7** (0.4) | 3.4 (7.2) | **2.5** (6.8) | - | - |
| | M→F | 16.8 | **41.3** | 17.6 | **66.7** | 61.5 (21.4) | **37.6** (10.4) | 83.4 (10.5) | **86.1** (9.3) | 3.8 (0.4) | **3.9** (0.4) | 2.7 (5.8) | **2.6** (6.2) | - | - |
| | F→F | 21.6 | **52.3** | 23.3 | **75.1** | 64.6 (25.1) | **40.7** (12.9) | 85.0 (9.8) | **86.4** (9.1) | 3.9 (0.4) | **4.0** (0.4) | 3.6 (8.9) | **2.6** (6.3) | - | - |
| | F→M | 23.0 | **54.1** | 22.6 | **62.6** | 68.6 (23.3) | **44.9** (14.0) | 74.6 (16.4) | **81.6** (11.9) | 3.6 (0.5) | **3.7** (0.5) | 5.4 (10.4) | **4.3** (9.7) | - | - |
| **VCTK → VCTK** | All | - | - | 31.1 | **39.9** | 51.7 (11.6) | **50.7** (11.7) | **74.2** (16.5) | 72.2 (19.4) | 3.2 (0.5) | 3.2 (0.5) | **3.1** (8.7) | 3.4 (8.6) | 22.7 (12.1) | **19.6** (11.4) |
| | M→M | - | - | 14.5 | **23.8** | 49.7 (11.6) | **49.1** (12.3) | **64.1** (16.3) | 60.4 (18.3) | 3.1 (0.5) | 3.1 (0.5) | **3.3** (9.1) | 3.8 (9.4) | - | - |
| | M→F | - | - | 43.8 | **62.5** | 49.0 (11.4) | **47.3** (11.5) | 82.6 (12.5) | **82.9** (12.4) | 3.6 (0.4) | 3.5 (0.4) | **3.5** (8.2) | 4.1 (9.0) | - | - |
| | F→F | - | - | 45.2 | **52.0** | **51.2** (9.8) | 52.5 (10.7) | 86.9 (8.5) | **87.1** (8.5) | 3.4 (0.4) | 3.3 (0.4) | 3.1 (9.1) | **2.9** (7.6) | - | - |
| | F→M | - | - | 30.7 | **34.2** | 56.3 (11.7) | **53.4** (11.1) | **70.0** (14.1) | 66.7 (19.2) | 3.0 (0.5) | 2.9 (0.5) | **2.6** (8.2) | 2.9 (8.0) | - | - |
| | English→English | - | - | 28.5 | **38.0** | 49.3 (11.5) | **47.9** (11.5) | **71.8** (16.9) | 70.2 (19.6) | 3.2 (0.5) | 3.1 (0.5) | 3.6 (9.7) | 3.6 (9.2) | - | - |
| | American→English | - | - | 34.3 | **44.2** | 52.8 (11.0) | **52.3** (11.5) | **76.9** (15.4) | 74.3 (18.9) | 3.2 (0.5) | 3.2 (0.5) | **2.5** (7.7) | 3.3 (8.2) | - | - |
| | Canadian→English | - | - | 29.9 | **35.3** | 54.3 (12.0) | **53.2** (11.5) | **73.8** (16.9) | 72.0 (19.5) | 3.1 (0.5) | **3.2** (0.5) | 3.5 (8.3) | **3.1** (8.0) | - | - |

## 4.4. Results and Discussion

The results of the objective evaluations are summarized in Table 2. For all conversions involving ESD and VCTK corpora, the proposed StarGAN-VC++ outperforms Baseline with respect to all emotion-preservation metrics. In terms of ACC$_{svm}$, Baseline only manages to achieve 25.9% emotion preservation accuracy, whereas StarGAN-VC++ achieves 52.7%. As per MAE$_{embed}$, StarGAN-VC++ achieves a significantly lower mean score of 45.7 compared to Baseline's score of 58.2. Also for pitch correlation, StarGAN-VC++ achieves a higher value of PCC (78.3) compared to Baseline (77.3). These results are statistically significant, as paired t-test achieves $p < 0.0001$ on PCC and MAE$_{embed}$ metrics.

For naturalness, both methods show similar pMOS values, which indicate that the proposed emotion-preservation techniques do not degrade the naturalness of the conversions. As far as content-preservation is concerned, StarGAN-VC++ shows a statistically significantly ($p < 0.05$) lower CER value (3.2%) than Baseline (3.5%). This improvement might be attributed to the proposed $\mathcal{L}_{inv}$ loss. The results also reveal that the speaker anonymization capability is not hampered by the introduction of emotion-preserving losses, as the mean SSS values for Baseline and StarGAN-VC++ are 24.4 and 23.9 respectively. This result is also statistically significant as $p < 0.001$ in paired t-test.

We also evaluate the models emotion-wise. For all 5 emotions in $e$, StarGAN-VC++ outperforms Baseline with respect to emotion preservation. Interestingly, StarGAN-VC++ shows significantly higher mean PCC values for all of these emotions compared to Baseline. This reveals that for emotional utterances, the variations of pitch are also better preserved by StarGAN-VC++, where pitch variations are indicative of emotional cues in speech [32]. The results also show that Baseline deals with *neutral* utterances better than other emotions, as it attains a very high score of 79.7% for ACC$_{svm}$, whereas for other emotions ACC$_{svm}$ lies in the range of 1.5% - 19.1%. Furthermore, it appears that *surprise* is the most difficult emotion to preserve, as both Baseline and StarGAN-VC++ models attain a 0% score for ACC$_{gt}$. However, in terms of ACC$_{svm}$, our StarGAN-VC++ model gets 62.9% accuracy against 1.5% by Baseline. As per [33], *surprise* is also the most difficult emotion for emotion recognition tasks as well.

With respect to gender-wise evaluation for ESD→ESD, StarGAN-VC++ shows a similar trend of better emotion-preservation as per the metrics. StarGAN-VC++ shows significant improvement in PCC values over Baseline and, also achieves a lower CER. StarGAN-VC++ model achieves the highest ACC$_{svm}$ value of 75.1% for F→F conversions against 23.3% by Baseline. However, in terms of ACC$_{gt}$, StarGAN-VC++ achieves the highest score of 54.1% for F→M conversions, against a score of 23.0% by Baseline. For VCTK→VCTK conversions, the improvement in PCC and MAE$_{embed}$ are not that significant. This is because the utterances in VCTK do not have emotional utterances with high pitch and intonation variations and, the change in F0 contour is relatively less compared to ESD utterances.

With respect to accent-wise evaluation, we assess for English→English, American→English, and Canadian→English sub-groups. The models were not trained with utterances from American and Canadian speakers from the VCTK dataset, which forms the unseen→seen speaker evaluation scenario as well. StarGAN-VC++ model outperforms Baseline in all three accent scenarios with respect to emotion preservation, scoring significantly higher ACC$_{svm}$ values and lower MAE$_{embed}$ values. Both Baseline and StarGAN-VC++ models achieve the highest ACC$_{svm}$ values for American→English conversions, 34.3 % and 44.2% respectively. StarGAN-VC++ does not manage to achieve higher PCC values for these accent-based conversion sub-groups.

We also perform an ablation study on the proposed losses and the results are presented in Table 3. The ablation study reveals that both of the proposed losses, deep embeddings-based and the augmented style reconstruction losses have a significant impact on emotion preservation. In the absence of $\mathcal{L}_{demo}$ loss, ACC$_{svm}$ value comes down to 36.4% from 52.7% for StarGAN-VC++. Similarly, when vanilla $\mathcal{L}_{style}$ is used instead on the proposed augmented version, the ACC$_{svm}$ value comes down to 34.3%. The drop is higher than that of $\mathcal{L}_{demo}$ loss, which implies that the augmented style reconstruction loss results in a better disentanglement of target speakers embeddings, and that facilitates emotion preservation. The use of augmented style reconstruction loss also improves the pitch correlation, as it achieves the mean PCC value of 79.8. In terms of quality, the pMOS value remains unchanged in all cases.

The results for the subjective evaluation are presented in Table 4. In terms of MOS, our proposed method does not degrade

---
[1]https://huggingface.co/speechbrain
[2]https://www.crowdee.com/

Table 3: *Ablation experiments results. Mean and standard deviation (in brackets) are reported.*

| Method | ACC$_{svm}$ [%] ↑ | PCC [×$10^2$] ↑ | pMOS ↑ | CER [%] ↓ |
|---|---|---|---|---|
| Baseline | 25.9 | 77.3 (15.3) | 3.5 (0.5) | 3.5 (8.5) |
| StarGAN-VC++ | **52.7** | **78.3** (16.7) | 3.5 (0.6) | **3.2** (8.0) |
| $\lambda_{demo}=0$ | **36.4** | **79.8** (14.0) | **3.5** (0.5) | 4.3 (9.5) |
| Vanilla $\mathcal{L}_{style}$ | 34.3 | 77.5 (15.3) | 3.4 (0.6) | 4.8 (9.9) |

the naturalness of the conversion, rather it is marginally better (4.1) than Baseline (3.9). The second column in Table 4 represents the number of times the converted sample's emotion is marked same as the source, i.e. no emotion leakage happened. StarGAN-VC++ achieves higher votes (331) than Baseline (258). The last column depicts whether the user marked the converted utterance's emotion similar to the reference rather than the source, another additional check for emotion leakage. For Baseline it is 209 votes, whereas for StarGAN-VC++ it is 114 votes, which further confirms that our proposed approach reduces the leakage. The audio samples are provided in the supplementary. The code can be found online [3].

Table 4: *Subjective evaluation results. Mean and standard deviation (in brackets) values are reported for MOS. StarGAN-VC++ is denoted by SG++. The source utterances received mean MOS score 4.0 (0.9).*

| MOS ↑ | | Conv. == Source Emo. ↑ | | Conv. == Ref. Emo. ↓ | |
|---|---|---|---|---|---|
| **Baseline** | **SG++** | **Baseline** | **SG++** | **Baseline** | **SG++** |
| 3.9 (1.0) | **4.1** (0.8) | 258 | **331** | 209 | **114** |

## 5. Conclusion

In this paper, we investigate the cause of failure of the state-of-the-art VC method StarGANv2-VC with respect to emotion preservation. Our study reveals that the failure is attributed to the framework's disability in the disentanglement of the target speaker's style and emotion embeddings while training. Consequently, the emotional content from the reference utterance leaks to the decoder, hampering the emotion preservation of the source utterance. We propose a novel deep emotion-embedding generation technique and emotion-aware losses for the VC, which encourages the generator to preserve source emotion. The objective and subjective evaluation results show that the proposed model improves the emotion preservation capability for diverse emotions, gender and accent groups, without compromising the quality of the converted samples. Further, the results show that for emotional data with high pitch variations, our proposed method improves the pitch correlation considerably, implying F0-consistent conversions. As a future task, we plan to use the proposed emotion-aware losses with other TTS-based methods. Also, we intend to generate emotion embeddings by using other representations of emotions, such as the two dimensional model of valence and arousal [34].

## 6. Acknowledgements

This research has been partly funded by the Federal Ministry of Education and Research of Germany in the project Emonymous (project number S21060A) and partly funded by the Volkswagen Foundation in the project AnonymPrevent (AI-based Improvement of Anonymity for Remote Assessment, Treatment and Prevention against Child Sexual Abuse).

---

[3]https://github.com/arnabdas8901/StarGAN-VC_PlusPlus.git

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
