# OpenReview forum: "StarGAN-VC++: Towards Emotion Preserving Voice Conversion Using Deep Embeddings"
_Interspeech.org/2023/Workshop/SSW — SSW12_

### Official Review · Reviewer_zjQV · 2023-05-30
**StarGAN-VC++: should be accepted**

**Rating:** 8
**Confidence:** 2

**Review:**

Key Strength of the paper: Generally well-written, technically sound, includes both objective and subjective evaluation, good summary of
related work.

Main Weakness of the paper: None.

Novelty/Originality, taking into account the relevance of the work for the SSW audience: Highly relevant.

Technical Correctness. The work appears to be technically solid. I am not sufficiently familiar with this line of research to assess if sufficient details are provided to reproduce the experimental work.

Suggestions for improvement: None, except the issue of illegible tables and figures (see below).

Quality of References: Adequate.

Clarity of Presentation: Generally clear. However, figures and tables are impossible to read when printed on paper. Even on the computer screen, table 2 is next to useless, because once the table is magnified to become legible, the overview of the table is lost.

---

### Official Review · Reviewer_4FJH · 2023-06-04
**Interesting approach for speech factorisation but unclear application and questions about the significance in the evaluation.**

**Rating:** 6
**Confidence:** 4

**Review:**

Strong points:

- The paper is well written and in general is easy to follow.
- The experimental results seem  reasonably good and the evaluation appears to be methodologically correct and complete, especially the objective metrics.

Weakness:
- The ultimate goal is not  clear. From the introduction, the addition of the adversarial speaker classifier loss and the evaluation metrics it seems that the main goal is speaker anonymization. In that  case, it’s not clear to me why a target reference speaker is needed instead of just trying to produce consistent speech in a non-identifiable voice. On the other hand, if the goal is a general VC system,  some speaker similarity metrics between the  target speaker and the output should be reported.
- Considering the standard deviations, many of the results do not seem to be significant. For example I find hard to see how the results for SSS 24.4 vs 23.9 with std of 15.9 and 16.6 can be statistically significant. Assuming  the given mean and std values and a 3300 sampling points (1800 ESD+1500 VCTK) the t score should be 0.5/(16.25*sqrt(2/3300)) =1.25 which is lower than even the right-tail  t-value for a 95% significance with 6600 degrees of freedom = 1.64, so the null hypothesis cannot be rejected
- The subjective part of the evaluation is significantly weaker than the objective one.

Novelty :
Not outstanding but sufficient.

Technical correctness:
See above comment on statistical significance values.

This paper deals with the problem of factorising speech into different components: content, style, speaker, etc, in the framework of voice conversion. In concrete, it introduces a proposal of voice conversion that preserves the emotion/style of the source speech while changing the speaker identity to that of a different speakers. The authors introduce several modifications on top of a base start-GAN architecture to  avoid leakages between  style/emotion and  speaker-identity information. The objective and subjective evaluation indicate that the proposed methods are reasonably effective.

Other suggestions:
- For the accent-wise evaluation, I think it should probably say “British” rather than “English”.
-  Calling Style encoder in both the speaker embedding of figure 1 and the emotion embedding of figure 3 is confusing
-  How are the Conversion Invariant Feature Preservation Loss different from feature-matching loss? I assume ’N’ here refers to the encoder. This should be made clear in the text.
- In section 2.1 you mention a separate mapping network M but it is not clear where in the model  this is used.
- Section 2.2: I’d be very careful in using any visual inspection of  a tSNE projection as prove of anything. See for example https://arxiv.org/pdf/2110.02573.pdf)

References:  Adequate.

Clarity of presentation: Good in general. See comments in "other suggestions".

---

### Decision · Program_Chairs · 2023-06-14

**Decision:**

Accept

**Comment:**

SSW2003 received 45 papers. The acceptance rate is 82%. We are pleased to inform you that your paper has been accepted by the SSW2023 Program Committee. Please read the reviews carefully and submit your camera-ready paper by June 28th. Most reviewers performed a detailed review. Please answer to their questions and consider their comments. Note that camera-ready papers are credited with one extra page to allow authors to consider reviewers’ suggestions. So max 7 pages in total including figures & refs.
The deadline for submitting the revised version (with full non-anonymized authors and refs!) is 28th June.